# Hybrid Raman and Laser-Induced Breakdown Spectroscopy for Food Authentication Applications

**DOI:** 10.3390/molecules28166087

**Published:** 2023-08-16

**Authors:** Sungho Shin, Iyll-Joon Doh, Kennedy Okeyo, Euiwon Bae, J. Paul Robinson, Bartek Rajwa

**Affiliations:** 1Department of Basic Medical Sciences, Purdue University, West Lafayette, IN 47907, USA; idoh@purdue.edu (I.-J.D.); wombat@purdue.edu (J.P.R.); 2Weldon School of Biomedical Engineering, Purdue University, West Lafayette, IN 47907, USA; kokeyo@purdue.edu; 3School of Mechanical Engineering, Purdue University, West Lafayette, IN 47907, USA; ebae@purdue.edu; 4Bindley Bioscience Center, Discovery Park, Purdue University, West Lafayette, IN 47907, USA

**Keywords:** laser-induced breakdown spectroscopy (LIBS), Raman spectroscopy, compact and combined system, food authentication, simultaneous atomic and molecular analysis

## Abstract

The issue of food fraud has become a significant global concern as it affects both the quality and safety of food products, ultimately resulting in the loss of customer trust and brand loyalty. To address this problem, we have developed an innovative approach that can tackle various types of food fraud, including adulteration, substitution, and dilution. Our methodology utilizes an integrated system that combines laser-induced breakdown spectroscopy (LIBS) and Raman spectroscopy. Although both techniques emerged as valuable tools for food analysis, they have until now been used separately, and their combined potential in food fraud has not been thoroughly tested. The aim of our study was to demonstrate the potential benefits of integrating Raman and LIBS modalities in a portable system for improved product classification and subsequent authentication. In pursuit of this objective, we designed and tested a compact, hybrid Raman/LIBS system, which exhibited distinct advantages over the individual modalities. Our findings illustrate that the combination of these two modalities can achieve higher accuracy in product classification, leading to more effective and reliable product authentication. Overall, our research highlights the potential of hybrid systems for practical applications in a variety of industries. The integration and design were mainly focused on the detection and characterization of both elemental and molecular elements in various food products. Two different sets of solid food samples (sixteen Alpine-style cheeses and seven brands of Arabica coffee beans) were chosen for the authentication analysis. Class detection and classification were accomplished through the use of multivariate feature selection and machine-learning procedures. The accuracy of classification was observed to improve by approximately 10% when utilizing the hybrid Raman/LIBS spectra, as opposed to the analysis of spectra from the individual methods. This clearly demonstrates that the hybrid system can significantly improve food authentication accuracy while maintaining the portability of the combined system. Thus, the successful implementation of a hybrid Raman-LIBS technique is expected to contribute to the development of novel portable devices for food authentication in food as well as other various industries.

## 1. Introduction

Food fraud has become a significant worldwide concern, as it often results in food adulteration and contamination. This can not only harm consumer trust and brand loyalty but can also lead to foodborne outbreaks due to the introduction of foodborne pathogens or the toxicity of added ingredients [1,2,3,4]. For this reason, various technologies have been adapted for the analysis of food samples, including vibrational spectroscopy and mass spectrometry techniques [5]. As adulteration in the food supply chain can happen at different stages, e.g., production, packaging, shipping, and distribution, it is crucial that the measurement technology be portable. Instrument portability can ensure immediate and effective detection and prevention of intentional or unintentional adulterations. Hence, new techniques for analyzing food on-site using optical sensors, like the Fourier transform infrared (FTIR) and compact photoacoustic laser systems, are constantly being developed [6,7]. These infrared (IR) techniques supply various molecular information to determine the authenticity and quality of foods. For instance, M. Pan et al. reported a hand-held FTIR device (TruDefender FTX, Thermo Fisher Scientific, Waltham, MA, USA) applied for the detection of adulterated olive oils [8].

Laser-induced breakdown spectroscopy (LIBS) and Raman spectroscopy systems have also emerged as promising technologies for food analysis because of their capability to perform real-time, multiplexed, and in-air measurements. Laser-induced breakdown spectroscopy (LIBS) is a non-destructive analytical technique that involves using intense laser pulses to generate plasma and measuring the resulting spectral intensity for elemental analysis. This method can be used for the qualitative analysis of various target materials [9,10]. A significant advantage of LIBS is that it is capable of real-time analysis of elemental composition with few restrictions. Raman spectroscopy measures the intensity of Raman scattering and uses it to provide a structural fingerprint, allowing for the identification of molecules [11,12]. Raman spectroscopy has the advantage of being non-destructive. Because of these advantages, handheld devices for both LIBS [13] and Raman spectroscopy [14] approaches are commercially available. However, these devices were typically developed for material science or pharmaceutical analysis.

Many researchers have shown that food can be authenticated using either LIBS or Raman spectroscopy. In particular, LIBS is gradually gaining popularity for real-time component analysis of various foods, for example, meat [15], milk [10], red wines [16], and fruits [5]. Raman spectroscopy has been used to carry out non-invasive analysis of beverages [17], dairy [18], oils [19], and cereals [20]. Although these techniques have produced satisfactory results, there is still a need for further improvement in their classification performance. For example, Zhang et al. [21] reported that a classification accuracy of over 80% was achieved using a support vector machine (SVM) to identify coffee varieties. Problems with relatively low classification accuracy for certain similar food groups were shown in our previous study. We reported that an average classification accuracy of about 85% was achieved with Alpine-style cheeses and coffee. In contrast, the classification of spices was possible with 95% accuracy [22].

Recently, a combination of LIBS and Raman systems has been demonstrated for the purpose of joint elemental and molecular analysis. These two optical methods share similar advantages, such as standoff detection, optical excitation, and potential for portability [23,24]. Yet, the application of such a combination for food analysis is still in its infancy. Miniaturizing instruments for handheld usage is limited by the complexity of their design. LIBS and Raman spectroscopy devices typically require an intensified charged coupled detector (ICCD)-based spectrometer, which has a large footprint and is better suited for a benchtop instrument [25,26]. In addition, the need for two distinct spectrometers for separate identification of the LIBS and Raman signals results in a significant increase in the cost and complexity of the system [27,28]. As a result, few studies have focused on food authentication applications. For example, Zhao et al. [29] reported that quantifying calcium in infant formula using LIBS and Raman required two spectrometers, a 1064 nm pulsed laser for LIBS and a 532 nm continuous laser for Raman. When it comes to combined LIBS and Raman systems, the focus remains mainly on mineral analysis because the power of the LIBS modality is leveraged to its fullest in this area [30,31]. Finally, data processing (chemometrics) is not currently optimized when handling combined systems in various analyses. Recent studies have utilized various data-fusion and feature selection methods to improve the performance of classification models [24,32,33].

For instance, Hoehse et al. merged spectra by aligning the X-axis, resulting in a uniform scale that allowed the LIBS and Raman spectra to be seamlessly integrated [28]. Zhao et al. [29] analyzed two different data fusion strategies involving concatenation and coaddition.

Here, we present the development and evaluation of a compact hybrid Raman/LIBS system (Hy-R-LIBS) and compare its performance in the context of food analysis with each conventional system. System validation was first performed using polystyrene (PS) beads, and the results were compared with data obtained from commercial equipment. Next, spectral analysis and classification were conducted using actual food samples. Specifically, the performance of two widely used classifiers was compared and analyzed when they were utilized with a multivariate feature selection approach, including two different data fusion methods.

We demonstrate that the elastic net (ENET) approach is the preferred technique for improving the classification performance when employing combined LIBS and Raman spectra. Our portable device has advanced detection capabilities, making it a promising tool for in-field food analysis. Further development and widespread use of compact combined LIBS/Raman detectors could lead to the emergence of new protocols for food product classification.

## 2. Results

### 2.1. System Validation

Figure 1 shows the normalized Raman (Figure 1a) and LIBS (Figure 1b) spectra from PS beads. Figure 1 compares spectra from the proposed instrument (solid line) and the corresponding reference instruments (dashed line). Ten single-shot data with ten different measurement locations (1 mm interval) on the target were averaged to generate Raman spectra from Hy-R-LIBS. As shown in Figure 1a, both instruments generated the Raman C-C breathing (984 cm^−1^), C-C stretch (1158 cm^−1^), and C=C stretch (1584 cm^−1^) bands in polystyrene (PS) bead samples [34]. The absence of a clearly distinguishable CH_2_ band in the Hy-R-LIBS spectrum can be attributed to the higher fluorescence background in our proposed system produced when utilizing shorter wavelengths of the excitation source [35]. Although the resolution and sensitivity of the commercial reference instrument were much higher owing to the sensitivity of the ICCD, the spectrum obtained with the compact CCD spectrometer displayed comparably similar bands, with only a minor spectral shift (2.3 cm^−1^ of averaged three peaks) attributable to the different excitation wavelength of the laser source.

Figure 1b reports the demonstration of LIBS capability. The molecular bands, such as CN (388.2 nm) and C_2_ band (swan band, such as 516.2 nm), were clearly observed for both Hy-R-LIBS and the commercially available handheld LIBS instrument. In addition, the spectral resolution and sensitivity of the two instruments were also comparable. These molecular peaks (CN and C_2_) are frequently detected in polymer samples using LIBS [36,37]. However, compared to the Raman instrument, the atomic spectrum of LIBS cannot identify the molecular structure of the target. It should be noted that an elemental peak of Na (588.9 nm) was also detected. This was an unexpected finding. However, a small amount of sodium borohydride might be present in the sample, as this substance is used to prevent oxidative degradation of the polymer when manufacturing the beads [38,39,40]. Furthermore, simultaneous detection was also performed using PS beads, as shown in Appendix A. It was observed that C_2_ (LIBS) and C-C peaks (Raman) overlapped at around the 562 nm range. It is important to note that the Na (588.9 nm) peak, potentially caused by air or contamination, needs to be subtracted from the line profile by using a Lorentzian function. This is necessary as the peak may overlap with the proposed Raman spectra in the range of about 1820 cm^−1^ [41].

### 2.2. LIBS Measurement Results

Figure 2 shows averaged LIBS spectra obtained from (Figure 2a) 16 Alpine-style cheeses and (Figure 2b) 7 Arabica coffee varieties. The spectra were all measured under the same conditions using the Hy-R-LIBS. A total of 100 spectra were averaged in these plots, and each single spectrum was used for the classification procedure. For easier visualization of the spectra, only three representative cheese sample data are shown. The full spectra for all 16 samples can be found in Appendix A. Note that the 16 different cheese samples used in this study can be classified into three general groups (Gruyère cheese (C11), US-manufactured Gruyère-style cheese (C16), and other Alpine-style cheese (C6)). CN band, Ca ionic, Ca atomic, C_2_ band, and Na atomic peaks were detected in all samples at relatively lower pulse energy compared to our previous study using a benchtop LIBS system (handling laser pulse energy of 62 mJ) [42]. These dominant peaks are visibly similar except for minor variations within food types or the same food group. It was also observed that elemental peaks such as Ca and Na were much higher in food samples than molecular bands obtained by LIBS. The same measurements were repeated using the commercial LIBS instrument mentioned above. The results of this measurement showing both normalized spectra are shown in Appendix A.

### 2.3. Raman Spectroscopy Results

The results of Raman spectroscopic measurements conducted on cheese samples and coffee varieties are shown in Figure 3a and Figure 3b, respectively. All the spectra shown are averages of 100 individual spectra obtained with our custom-built instrument. Figure 3a shows that distinctive Raman peaks could be detected depending on the food product types, while only minor differences were noticed within the same food group. Because cheese contains lipids and protein, several Raman bands are typically observed [43]. These bands can be different or shifted under different processing conditions, such as different additives. Hence, several fat-related bands were observed using Hy-R-LIBS, as shown in Figure 3a. The 890 cm^−1^ band is supposed to be a phospholipid headgroup. The 1287 cm^−1^ band is likely to be caused by CH_2_ twisting in the phospholipids, while the 1432 cm^−1^ band can be attributed to CH_2_ scissoring from cholesterol. The band at 1670 cm^−1^ is typically due to C=C stretching in the phospholipids [44]. However, protein bands could not be detected because of the limited detection range of the spectrometer used in the proposed system.

Next, coffee varieties were analyzed, and as shown in Figure 3b, three dominant Raman bands can be clearly observed. These bands were most commonly present in the Arabica samples and can be attributed to the aromatic and phenolic acids, which are important constituents of coffee [45]. The band at about 1440 cm^−1^ is caused by CH_3_ deformation vibrations [45,46]. The bands at about 1570 and 1650 cm^−1^ correspond to the carbon C=C stretching of 1,3-cyclohexadiene and cyclohexene, respectively [46,47]. For comparison, an additional Raman test was repeated using a commercial instrument. The Raman measurement results for the cheese and coffee varieties shown in Appendix A, respectively, demonstrate the same bands as observed in Figure 3.

### 2.4. Classification Results

Figure 4 summarizes the classification results of 16 cheese samples and 7 coffee varieties using a multinomial logistic regression model with an elastic net (ENET) regularizer. Three different input settings, including individually performed Raman and LIBS spectra and combined spectra, were compared. It was demonstrated that the classification accuracy for both cheeses and coffee could be significantly improved by at least 5% when the combined LIBS and Raman spectra were used. For example, the average classification accuracy achieved was 94.34% when combined LIBS and Raman data from coffee varieties were used, compared with only 85.17% obtained with LIBS alone. One of the confusion matrices for all cheese and coffee samples is shown in Appendix A. In addition, a proposed fusion 2 (coaddition) method using both LIBS and Raman spectra showed about a 1–2% increase in classification accuracy compared with the simple fusion (concatenation) method. This implies that an appropriate feature selection resulting in fewer features is necessary for the ENET, along with a support vector machine (SVM)-based classifier to help reduce noise and effects of overfitting. For instance, the number of input variables after ENET used with concatenation and coaddition were 155 and 104, respectively. The total number of selected features and classification results are summarized in Table 1 and Table 2.

The SVM classifier was selected as the basic benchmark in this study. The classification results of cheese samples and coffee varieties by SVM are summarized in Table 1 and Table 2, respectively. As with the ENET classifier, combined LIBS and Raman spectra could enhance classification accuracies from the SVM classifier. It was also shown that the ENET classifier showed slightly better classification performance than the SVM classifier employing ENET-based feature selection.

## 3. Discussion

This system is a prototype designed for future commercialization. The current dimensions of the chassis that contain all the system components (including two lasers) are 15 × 10 × 5 cm. Similarly, Alvarez-Llamas et al. [48] reported a system in which all laser and optical components were mounted in a single module with an overall dimension of about 25 × 10 × 5 cm. This makes it possible for the instrument to be configured into a portable device. The overall size can be further reduced if the employed diode-pumped solid-state (DPSS) laser were smaller. A single pulsed laser operating at 532 nm can be used to excite both Raman and LIBS signals. However, owing to the need for a highly sensitive detector such as an ICCD spectrometer, the use of a pulsed laser for Raman analysis is expected to be limited in a portable system. Matroodi et al. [49] reported the simultaneous recording of Raman and LIBS achieved with 5- and 35-mJ pulses generated by a single laser. However, the signal was acquired using an ICCD.

The classification performance results are summarized in Figure 4. Interestingly, similar single-method classification accuracies of about 85% in both the cheese and the coffee were measured using LIBS spectra only in our previous study [42], even though that system had higher laser output (e.g., a pulse energy of 62 mJ within the same spectral range of 350–600 nm).

It was anticipated that relying solely on the LIBS technique would have limitations in identifying food samples. The same was observed with Raman spectroscopy when used alone. The classification performance using Raman spectra was only about 80%, which was lower by 5% compared to the results from LIBS. This subpar classification performance may be attributed to the relatively narrow Raman spectral range as compared to that of LIBS. To improve the classification of cheeses by Raman, it is highly probable that a broader spectral range would be beneficial. This is because there is a major band associated with lipid vibration that is likely to occur around 2900 cm^−1^ [50].

The ENET method was employed for the classification of two food-product categories. The ENET approach has already been demonstrated to show both excellent feature selection and classification performance in various spectroscopic research studies [51,52]. The accuracy of classification based on the proposed ENET classifier increased by about 10% for both samples when employing the hybrid Raman and LIBS systems. The SVM classifier yielded similar, slightly inferior results (see Table 1 and Table 2). These results showed that combining both elemental and molecular information can significantly improve the classification performance in foods. Consequently, such a combination in a single platform could be a critical improvement in portable optical-based food fraud detection. Our results agree with a report of Hoehse et al., who also reported that the predicted classification of pigments was enhanced by merging the LIBS and Raman datasets [28].

The impact of the data fusion method was also investigated in this study. Two simple conventional data-assembling strategies were employed (fusion 1: concatenation; fusion 2: coaddition) [53,54]. Note that these two methods selected a different number of features when paired with ENET, as shown in Table 1 and Table 2. The fusion 2 method yielded higher classification performances when the number of features used in the classifier was lower in this study. Similarly, Zhao et al. [29] reported that the results achieved by coaddition (283 spectral variables) were slightly more accurate than those achieved by the concatenation (828 spectral variables) in a partial least squares regression (PLSR) model developed using both Raman and FT-IR spectra. According to J. Moros et al. [53], combining LIBS-Raman data through coaddition yields a simpler output that results in more successful classification compared to using the concatenation method. Therefore, the coaddition of Raman and LIBS datasets combined with the ENET approach provides the best option for the classification of the targets.

## 4. Materials and Methods

### 4.1. Sample Preparation

Polystyrene beads (PS, No. 441147, Sigma-Aldrich, St. Louis, MO, USA) were selected as a reference material for the validation test comparing the proposed system and commercial instruments in this study. The average molecular weight of PS is about 350,000.

Two different solid food examples were chosen for the analysis of food authentication: 16 Alpine-style cheese samples (representing from Ch1 to Ch16 in this study) and 7 commercially available Arabica coffee brands (representing from C1 to C7 in this study). The aforementioned food categories exhibited the least accurate classification rate in our prior investigation using LIBS; therefore, they were designated for the follow-up study. Sample sources and information pertaining to the cheese and coffee samples can be found in our previous manuscript. We encourage interested readers to refer to this publication for further information [42].

### 4.2. System Description

Figure 5 shows a schematic of Hy-R-LIBS consisting of a delay generator (DG), a mirror (M), a dichroic mirror (DM), a focusing lens (FL), a collection lens (CL), a notch filter (NF), an optical fiber (OF), and a motorized 3-axis stage (XYZ). This system also contained a pulsed laser (laser 1; MicroJewel DPSS laser, Quantum Composers, Bozeman, MT, USA) for LIBS, a continuous wave (CW) laser (laser 2; CP532, Thorlabs, Newton, NJ, USA) for Raman, and a visible (VIS)-range spectrometer (VIS; Avaspec Mini, Avantes, Apeldoorn, The Netherlands) for the detection of both LIBS and Raman. The pulsed laser had a pulse width of 6 ns and a pulse energy of 10 mJ, while the CW laser had a power of 5 mW. The theoretical laser beam spot size of the pulsed and CW lasers at the focal point was approximately 50 µm and 10 µm, respectively. A compact spectrometer provided a spectrum in the 350–625 nm range with 0.33 nm of spectral resolution. The gate width of the spectrometer for LIBS was set to 1.05 ms with a 1.0 µs gate delay, and the exposure time for Raman was chosen as 0.5 sec. Two collection lenses (with a focal length of 50 mm and an f-number of 0.5) were positioned at an approximately 45° angle from the incoming laser beam’s direction and linked using an optical fiber with a core diameter of 600 μm (FC-UVIR200-2, Avantes).

Both sequential and simultaneous Hy-R-LIBS spectra can be detected in a single compact spectrometer. Specifically, for sequential Hy-R-LIBS, the pulsed laser was turned on to generate a plasma emission signal after the CW laser generated Raman scattering within a specific acquisition time. For example, LIBS measurement within a gate width (1.05 ms) was performed after 1 s while completing the Raman signal for 500 ms at the same focal spots, and both measurements were repeated at different spots in the target. For simultaneous Hy-R-LIBS, the LIBS emission signal was generated while Raman scattering was continuously generated by the CW laser. It should be noted that LIBS and Raman’s results overlapped in the 540–625 nm spectral region, as shown in Appendix A. A specific dichroic mirror that transmits the Near-infrared (NIR) laser but reflects the VIS laser to the target was used to match the same focal spot. Both LIBS and Raman signals from the target were collected through shared collection optics into a single spectrometer where LIBS spectra were registered within the VIS range (350–625 nm), and Raman spectra were acquired within a range of 750–2800 cm^−1^ (about 555–625 nm spectral range). The notch filter blocks the CW laser to prohibit direct reflection from the CW laser source.

A commercially available handheld LIBS and a benchtop Raman system were selected as reference instruments. The handheld-LIBS (Z-900, SciAps Inc., Woburn, MA, USA) consisted of a 1064 nm laser (pulse energy of 5 mJ and pulse width of 1–2 ns), a spectrometer (spectral range of 190–900 nm), and optical assembly in a handpiece enclosure [55]. A single LIBS spectrum was measured for 1 ms after a 650 ns gate delay. The Raman spectroscopy system (Alpha300, WiTec, Ulm, Germany) consisted of a 635 nm laser (power of 15 mW) and an ICCD spectrometer (iDus 401, Andor, Belfast, UK) [56]. A single Raman spectrum was acquired during 0.5 sec exposure time.

For each food product, a total of 100 LIBS and 100 Raman spectra were sequentially collected at 25 different spots in four physically different specimens using a raster area of approximately 4 × 4 mm. The LIBS measurement was performed right after the Raman measurement before changing laser spots. For example, a total of 1600 LIBS spectra and 1600 Raman spectra were collected within 16 cheese samples.

### 4.3. Classification Methods

Figure 6 describes the overall data processing for the classification using separate Raman or LIBS signals or two different data fusion methods for LIBS and Raman signals before multivariate feature selection. First, raw Raman spectra were processed by elimination of the estimated baseline. Second, denoising, normalization, and transformation were conducted in all collected LIBS and Raman data to reduce the plasma fluctuation effects [57].

Several steps of feature selection were conducted before building a classifier. An analysis of variance (ANOVA) was selected as a method of univariate filtering to remove the features associated with very small effect sizes [58], and a regularized multinomial logistic regression model with elastic net (ENET) was used to perform multivariate feature selection while constructing the classification model [59,60].

Briefly, we define the ENET model as follows:$$\underset{\beta_{k}}{argmin}\left\{-\frac{1}{n}\sum_{i=1}^{n}\sum_{k=1}^{K}1\left(y_{i}=k\right)\log\left(\frac{\exp\left(x_{i}^{T}\beta_{k}\right)}{\sum_{l=1}^{K}\exp\left(x_{i}^{T}\beta_{l}\right)}\right)+\lambda \left[\alpha {∥\beta_{k}∥}_{1}+\frac{1}{2}(1-\alpha ){∥\beta_{k}∥}_{2}^{2}\right]\right\}$$
where $\beta_{k}$ is a vector of coefficients, $1(y_{i}=k)$ is an indicator function that returns 1 when the class *y* is equal to *k_j_*, and 0 otherwise; *x_i_* is the vector of predictors (the spectral features) for *i*-th observation; *λ* and *α* is the regularization parameter controlling the balance between ℓ_1_ (LASSO) and ℓ_2_ (Ridge) regularizations.

Incorporating both ℓ_1_ and ℓ_2_ regularization terms in the model produces effective feature selection. Specifically, the ℓ_1_ term facilitates the elimination of irrelevant or less crucial features by inducing coefficients to become zero. Meanwhile, the ℓ_2_ term penalizes larger coefficient values to prevent overfitting and improve the stability of the model. The ENET regression model was trained by tuning the model parameters (α and λ ≥ 0 and performing repeated 10-fold cross-validation. In addition, two different data fusion approaches for the LIBS and Raman spectra (concatenation (fusion 1) and coaddition (fusion 2), as shown in Figure 6), were compared. Note that the number of variables produced by the fusion 1 and the fusion 2 techniques after feature selection could be different. By distinguishing between more useful and less useful features, the fusion 2 step may result in a smaller number of variables being produced [61]. Detailed information about the variable numbers after feature selection for each condition is provided in Table 1 and Table 2.

Following the univariate feature selection method, two different classifiers, ENET and SVM, which are widely used in analysis [62], were executed and compared. Ten different training and testing sessions were conducted with distinct random seeds to evaluate performance variability. The sessions were conducted independently to ensure unbiased results. Finally, the mean and standard deviation from the diagonal value of a cross-validation matrix were computed to represent the results. All data processing steps reported in Figure 6 were developed and implemented using custom Matlab and Python scripts.

## 5. Conclusions

Combating food fraud on a global scale is a challenge that demands a multidisciplinary approach. The integration of various methods is crucial for enhancing the reliability and efficacy of food safety measures. The outcomes of the study indicate that the Hy-R-LIBS system, coupled with its chemometric strategies, might be employed for food authentication with promising results.

Using an ENET-based classifier that integrates feature selection and downstream classification can effectively process both elemental and molecular data on a single platform. The implementation we reported and the resulting classification performance provide substantial support for the use of hybrid spectroscopic methods for food classification and, by extension, the detection of accidental contamination or outright food fraud.

Whether the methods presented are ready to be implemented for commercial use based on the achieved classification accuracies is a matter of debate. While a desirable accuracy should ideally approach 100%, practical screening protocols are multilayered, with early notification systems employed to identify possible candidates for further analysis using orthogonal bench-top methods, such as mass spectroscopy. Furthermore, in real-life situations, classifiers are optimized for specificity or sensitivity depending on the scenario and the cost of false negatives and false positives. The field of clinical diagnostics, where systems are typically optimized either for specificity or sensitivity, faces similar dilemmas. Naturally, this leads to a discussion of whether the positive and negative predictive values (PPV and NPV) should be used as the better metric. However, PPV and NPV are not just dependent on sensitivity and specificity but also on the prevalence of the tested problem. Therefore, the level of overall accuracy demonstrated by our prototype cannot determine whether the methodology outlined here is mature enough for immediate use in the field. The answer ultimately depends on individual use cases, as they may come with varying costs related to misclassification and different prevalence of mislabeling or adulteration. Hence, further evaluation is necessary to determine the practicality of the system.

## Figures and Tables

**Figure 1 molecules-28-06087-f001:**
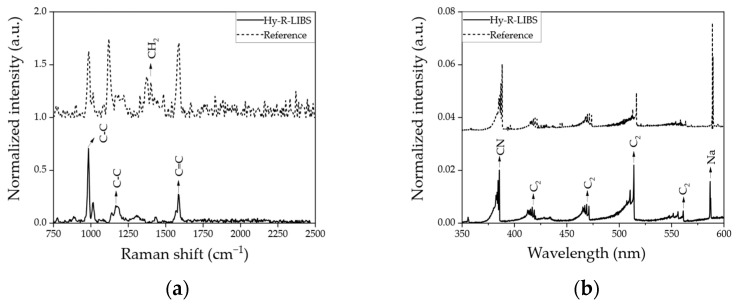
Normalized averages of ten spectra of PS beads obtained using (**a**) Raman and (**b**) LIBS systems. The solid and dashed lines represent the spectra acquired with the proposed and commercial instruments, respectively.

**Figure 2 molecules-28-06087-f002:**
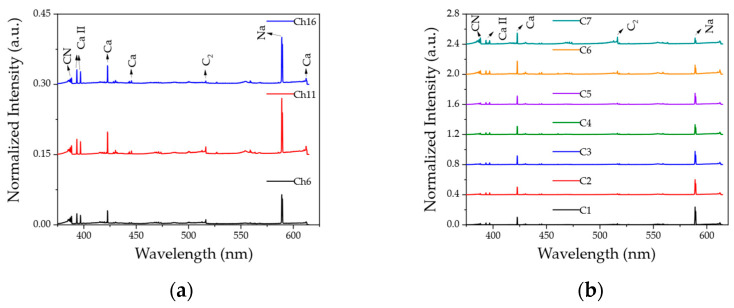
Averaged and normalized LIBS spectra for (**a**) 16 cheese samples and (**b**) 7 coffee varieties. Note that the data for all cheese samples are given in Appendix A.

**Figure 3 molecules-28-06087-f003:**
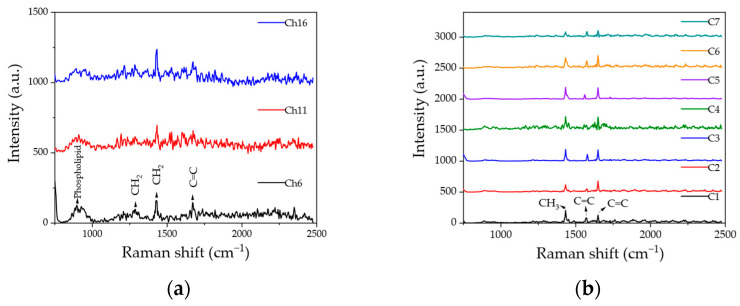
Averaged Raman spectra for (**a**) three representative cheese samples and (**b**) seven coffee varieties. Note that the data for all cheese samples are given in Appendix A.

**Figure 4 molecules-28-06087-f004:**
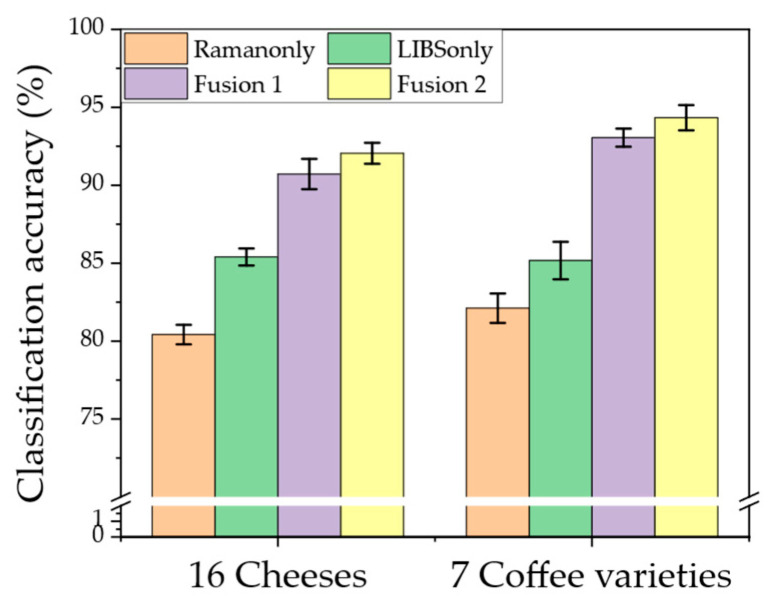
Classification plots using ENET classifiers for 16 cheeses and 7 coffee varieties from four data sets.

**Figure 5 molecules-28-06087-f005:**
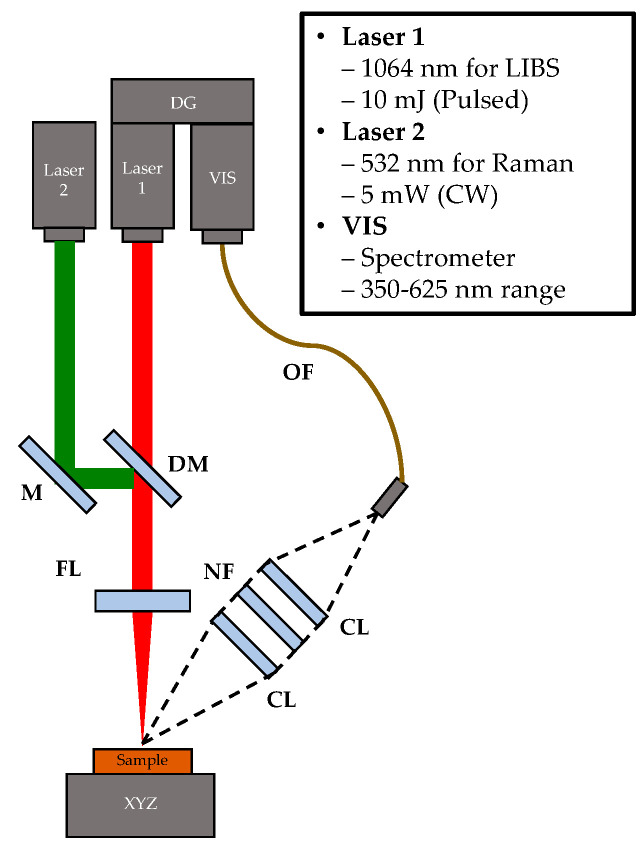
A schematic diagram of the Hy-R-LIBS system. The proposed system includes Laser 1 and Laser 2, a mirror (M), dichroic mirror (DM), focal lens (FL), two collection lenses (CLs), notch filter (NF), optical fiber (OF), visible range spectrometer (VIS), and delay generator (DG).

**Figure 6 molecules-28-06087-f006:**
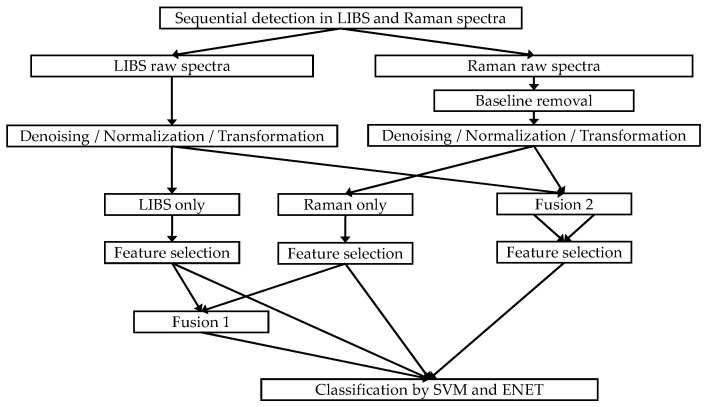
Diagram of data processing.

**Table 1 molecules-28-06087-t001:** The total number of selected features after an analysis of variance (ANOVA) and ENET and classification accuracies from two different classifiers using cheese samples. Note that *v*, *n*, and % indicate the number of input variables, number of selected features, and classification accuracy, respectively. The mean and standard deviation of the diagonal of the cross-validation matrix were also computed in this study.

Method	Raman (*v* = 600)	LIBS (*v* = 2000)	Fusion 1	Fusion 2 (*v* = 2600)
ANOVA (*n*)	200	200	-	200
ENET (*n*)	57	98	155	104
SVM (%)	79.67 (0.90)	84.66 (0.42)	90.57 (1.01)	91.48 (0.70)
ENET (%)	80.42 (0.63)	85.40 (0.55)	90.72 (0.97)	92.05 (0.67)

**Table 2 molecules-28-06087-t002:** The total number of selected features after ANOVA and ENET and classification accuracies from two different classifiers using coffee varieties.

Method	Raman (*v* = 600)	LIBS (*v* = 2000)	Fusion 1	Fusion 2 (*v* = 2600)
ANOVA (*n*)	200	200	-	200
ENET (*n*)	81	46	127	76
SVM (%)	82.74 (0.64)	85.15 (0.64)	92.28 (0.56)	94.20 (0.70)
ENET (%)	82.11 (0.94)	85.17 (1.20)	93.06 (0.58)	94.34 (0.81)

## Data Availability

The data presented in this study are available upon request from the corresponding author, subject to a confidentiality agreement.

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
