# Peer review of "Hybrid Raman and Laser-Induced Breakdown Spectroscopy for Food Authentication Applications"

_molecules, 2023, doi:10.3390/molecules28166087_

Round 1
Reviewer 1 Report
The manuscript is devoted to the development and testing of a system based on the combined use of Raman and LIBS spectroscopy for food analysis. The work looks interesting and useful. The text is very clear to the reader. This work can be recommended to publication after minor improvements.
1. Fig. 1a. The authors should explain why the CH2 peak is present in the reference spectrum and absent in the HYRLIBS spectrum.
2. Fig. 1b. I have doubts about the presented identification of the bands (CN and C2). Please check it.
3. L. 248. Give more details about ENET.
4. Fig. 5. Add a description of the abbreviations from the figure in the caption.
5. L. 287-299. What is the diameter of the optical fiber used? What are the parameters of the collection optics (focus and f-number)?
Author Response
We appreciate the comments and critiques from the reviewer. We revised the manuscript following the reviewer’s suggestions. The detailed response to the points raised by the reviewer is attached below.
The manuscript is devoted to the development and testing of a system based on the combined use of Raman and LIBS spectroscopy for food analysis. The work looks interesting and useful. The text is very clear to the reader. This work can be recommended to publication after minor improvements.
- Fig. 1a. The authors should explain why the CH2 peak is present in the reference spectrum and absent in the HYRLIBS spectrum.
We hypothesize that the high fluorescence background decreases the relative height of specific peaks, such as CH2 peak detected in ours and another study [34]. Besides the higher sensitivity obtained with the use of ICCD in the reference spectrum, the excitation wavelength of 633 nm (reference system) has also lower fluorescence background [35] than that of 532 nm (proposed system). The description has been added as per the reviewer’s suggestion as follows:
“The absence of a clearly distinguishable CH2 band in the Hy-R-LIBS spectrum can be attributed to the higher fluorescence background in our proposed system produced when utilizing shorter wavelengths of excitation source [35].”
- Fig. 1b. I have doubts about the presented identification of the bands (CN and C2). Please check it.
Based on the reviewer's suggestion, we conducted a further literature review to confirm the apparent detection of CN and C2 bands in polymer samples using LIBS. An additional reference by Weidman et al. [37] has been added in the result section to support the presented identification of the bands (CN and C2). According to Weidman and colleagues, an increased molecular emission of C2 and CN was detected during the examination of a polystyrene film with a double-pulse LIBS system.
- L. 248. Give more details about ENET.
A brief description of the ENET (elastic net regularization) method for classification and feature selection has been added to the manuscript, as suggested by the reviewer.
- Fig. 5. Add a description of the abbreviations from the figure in the caption.
The caption of Figure 5 has been modified as per the reviewer’s recommendation as follows:
“The proposed system includes Laser 1 and Laser 2, a mirror (M), dichroic mirror (DM), focal lens (FL), two collection lenses (CL), notch filter (NF), optical fiber (OF), visible range spectrometer (VIS), and delay generator (DG).”
- L. 287-299. What is the diameter of the optical fiber used? What are the parameters of the collection optics (focus and f-number)?
The characteristics of optical fiber and collection optics were added in section 4.2 as follows:
“Two collection lenses (with a focal length of 50 mm and an f-number of 0.5) were positioned at an approximately 45° angle from the incoming laser beam's direction and linked using an optical fiber with a core diameter of 600 μm (FC-UVIR200-2, Avantes).”
Reviewer 2 Report
The paper deals with the demonstration of combined LIBS-Raman classification of food in two selected case studies, addressing interest for the dedicated compact handheld instrument which has been realized. The paper is original and well written, nevertheless a few minor improvements are needed to make clearer the back-ground and progress shown by the present results.
As far as the background is concerned, an explicit mention is required to different portable IR techniques, such as FTIR and the recent IR excitation with photoacoustic detection [see for instance: Fiorani L., Artuso F., Giardina I., Lai A., Mannori S., Puiu A. Photoacoustic Laser System for Food Fraud Detection Sensors 21 (ISSN 1424-3210), paper 4178 - 11 p. (2021)] supplying the same information on molecular groups as Raman, but with higher sensitivity and lower limitation on sample transparency.
In the introduction LIBS capabilities of both qualitative and quantitative analysis is reported (line 55), without stressing that only the former are relevant to the current classification proposed. Quantitative analysis might be of importance for fraud detection in case of substances present at trace level (e.g. heavy metal contamination), but it may present complex problems, due to the matrix effect that requires the availability of suitable reference samples for calibration or training of the data analysis program. A comment on this point is required, alternatively "quantitative" LIBS should not be considered.
In the classification results by data fusion the better performances of coaddition with respect to concatenation is reported (1 to 2% improvement). The possibility to obtain directly fused data ready for the analysis by the HYRLIBS prototype is reported in sect. 4.2, however in the end of this section it is stated that "For this study, the sequential detection mode was considered for the classification" (line 312-313). This is confusing the reader that cannot understand how the reported improvements with data fusion2 could be observed.
Finally the reported 10% improvement in classification, obtained when using the combined spectroscopic platform, does not seem enough to justify the efforts of further developping it for commercialization. A comment on this point is necessary in the conclusion, for instance stressing the pratical meaning of the current result or the perspectives of its further improvements.
Author Response
We appreciate the comments and critiques from the reviewer. We revised the manuscript following the reviewer’s suggestions. The detailed response to the points raised by the reviewer is attached below.
The paper deals with the demonstration of combined LIBS-Raman classification of food in two selected case studies, addressing interest for the dedicated compact handheld instrument which has been realized. The paper is original and well written, nevertheless a few minor improvements are needed to make clearer the back-ground and progress shown by the present results.
- As far as the background is concerned, an explicit mention is required to different portable IR techniques, such as FTIR and the recent IR excitation with photoacoustic detection [see for instance: Fiorani L., Artuso F., Giardina I., Lai A., Mannori S., Puiu A. Photoacoustic Laser System for Food Fraud Detection Sensors21 (ISSN 1424-3210), paper 4178 - 11 p. (2021)] supplying the same information on molecular groups as Raman, but with higher sensitivity and lower limitation on sample transparency.
Following the reviewer’s recommendations, we explicitly mention the modern IR techniques in the introduction section as follows:
“Instrument portability can ensure immediate and effective detection and prevention of intentional or unintentional adulterations. Hence, new techniques for analyzing food on-site using optical sensors like the Fourier transform infrared (FTIR) and compact photoacoustic laser systems are constantly being developed [6, 7] . These infrared (IR) techniques supply various molecular information to determine the authenticity and quality of foods. For instance, M. Pan et al. reported a hand-held FTIR device (TruDefender FTX, Thermo Fisher Scientific, Waltham, MA) applied for the detection of adulterated olive oils [8].”
- In the introduction LIBS capabilities of both qualitative and quantitative analysis is reported (line 55), without stressing that only the former are relevant to the current classification proposed. Quantitative analysis might be of importance for fraud detection in case of substances present at trace level (e.g. heavy metal contamination), but it may present complex problems, due to the matrix effect that requires the availability of suitable reference samples for calibration or training of the data analysis program. A comment on this point is required, alternatively "quantitative" LIBS should not be considered.
Based on the reviewer’s comment, we have decided not to emphasize “quantitative” LIBS in the introduction. The quantitative analysis using suitable reference samples will be addressed in our upcoming research reports.
- In the classification results by data fusion the better performances of coaddition with respect to concatenation is reported (1 to 2% improvement). The possibility to obtain directly fused data ready for the analysis by the HYRLIBS prototype is reported in sect. 4.2, however in the end of this section it is stated that "For this study, the sequential detection mode was considered for the classification" (line 312-313). This is confusing the reader that cannot understand how the reported improvements with data fusion2 could be observed.
The coaddition data set showed indeed an improvement. The original phrasing was a result of careless editing. As per the reviewer’s recommendation, removed the confusing statement.
- Finally the reported 10% improvement in classification, obtained when using thecombined spectroscopic platform, does not seem enough to justify the efforts of further developing it for commercialization. A comment on this point is necessary in the conclusion, for instance stressing the practical meaning of the current result or the perspectives of its further improvements.
The adequacy of the reported accuracies depends on specific case scenarios. It is a complex matter as the usefulness of any diagnostic test, screen, or assay is based on the cost of misclassification and the estimated prevalence of the screened issue. To address reviewer concerns, we included a section in the conclusion:
“Whether the methods presented are ready to be implemented for commercial use based on the achieved classification accuracies is a matter of debate. While a desirable accuracy should ideally approach 100%, practical screening protocols are multilayered, with early notification systems employed to identify possible candidates for further analysis using orthogonal bench-top methods, such as mass spectroscopy. Furthermore, in real-life situations, classifiers are optimized for specificity or sensitivity depending on the scenario and the cost of false negatives and false positives. The field of clinical diagnostics where systems are typically optimized either for specificity or sensitivity, faces similar dilemmas. Naturally, this leads to a discussion of whether the positive and negative predictive values (PPV and NPV) should be used as the better metric. However, PPV and NPV values are not just dependent on sensitivity and specificity but also on the prevalence of the tested problem. Therefore, the level of overall accuracy demonstrated by our prototype cannot determine whether the methodology outlined here is mature enough for immediate use in the field. The answer ultimately depends on individual use cases, as they may come with varying costs related to misclassification and different prevalence of mislabeling or adulteration. Hence, further evaluation is necessary to determine the practicality of the system.”
Reviewer 3 Report
Minor corrections of methods and text are required:
The paper was written fine, the only additional specific comment could be Justification of proper, above-threshold fluences for LIBS — both for cheeses and coffees, to avoid LIBS intensity instabilities.
Controls for LIBS and Raman spectroscopy should be provided (e.g., sodium is present eerywhere, e.g., in air, how it is subtracted etc.)
Double check for awkward phrases - see, e.g., "For instance, Hoehse et al. [25] concatenated the LIBS spectra and the Raman spectra to create a uniform spectral x-axis within combined data." (1D data array?)
Minor editing of English language required
Author Response
We appreciate the comments and critiques from the reviewer. We revised the manuscript following the reviewer’s suggestions. The detailed response to the points raised by the reviewer is attached below.
The paper was written fine, the only additional specific comment could be Justification of proper, above-threshold fluences for LIBS — both for cheeses and coffees, to avoid LIBS intensity instabilities.
Thank you for your specific comment on avoiding LIBS instabilities. Before designing our proposed system in this study, we analyzed the LIBS signal and classification performances by comparing the benchtop system and commercial handheld LIBS unit. Although signal stability can be enhanced through the use of buffer gas or confinement, we opted for the most straightforward approach of enhancing pulse energy in our prototype. Our ultimate goal is to enable the detection of LIBS and Raman in various food matrices using a small-footprint instrument. We found that some liquid targets, such as vanilla extract, require additional drying to achieve a relatively stable LIBS signal. We are currently working on developing possible additional remedies to address the instabilities associated with such samples.
- Controls for LIBS and Raman spectroscopy should be provided (e.g., sodium is present everywhere, e.g., in air, how it is subtracted etc.).
According to the data in the supplementary Figure S1, in our system, the presence of sodium peak could be masked by other potential Raman spectra. For example, the sodium emission line (589 nm) due to LIBS may appear in the 1820 cm-1 range in our Raman readout. A description of how to subtract sodium peak was added in the result section:
“It is important to note that the Na (588.9 nm) peak, potentially caused by air or contamination, needs to be subtracted from the line profile by using a Lorentzian function. This is necessary as the peak may overlap with the proposed Raman spectra in the range of about 1820 cm-1.”
- Double check for awkward phrases - see, e.g., "For instance, Hoehse et al. [25] concatenated the LIBS spectra and the Raman spectra to create a uniform spectral x-axis within combined data." (1D data array?).
We changed the sentence to: “For instance, Hoehse et al. merged the spectra by aligning the X-axis, resulting in a uniform scale that allowed the LIBS and Raman spectra to be seamlessly integrated [28].”